# Multifidelity Reinforcement Learning with Control Variates

## Abstract

In many computational science and engineering applications, the output of a system of interest corresponding to a given input can be queried at different levels of fidelity with different costs. Typically, low-fidelity data is cheap and abundant, while high-fidelity data is expensive and scarce. In this work we study the reinforcement learning (RL) problem in the presence of multiple environments with different levels of fidelity for a given control task. We focus on improving the RL agent's performance with multifidelity data. Specifically, a multifidelity estimator that exploits the cross-correlations between the low- and high-fidelity returns is proposed to reduce the variance in the estimation of the state-action value function. The proposed estimator, which is based on the method of control variates, is used to design a multifidelity Monte Carlo RL (`MFMCRL`) algorithm that improves the learning of the agent in the high-fidelity environment. The impacts of variance reduction on policy evaluation and policy improvement are theoretically analyzed by using probability bounds. Our theoretical analysis and numerical experiments demonstrate that for a finite budget of high-fidelity data samples, our proposed `MFMCRL` agent attains superior performance compared with that of a standard RL agent that uses only the high-fidelity environment data for learning the optimal policy.

## 1   Introduction

Within the computational science and engineering (CSE) community, multifidelity data refers to data that comes from different sources with different levels of fidelity. The criteria by which data is considered to be low fidelity or high fidelity vary across different applications, but usually low-fidelity data is much cheaper to generate than high-fidelity data under some cost metric. In robotics for instance, data coming from a robot operating in the real world constitutes high-fidelity data, while simulated data of the robot based on first principles is considered to be low-fidelity data. Different simulators of the robot can also be designed by increasing the modeling complexity. A simulator that takes into account aerodynamic drag is, for instance, of higher fidelity than one that is based only on the simple laws of motion. As another example, a neural classifier in deep learning can be trained on the *full* training data for a *large* number of training epochs, or on a *subset* of the training data for *few* epochs. Evaluating the trained model on a held-out validation data set in the former case yields a higher-fidelity estimate of the classifiers' performance compared with that in the latter case. In general, low-fidelity data serves as an approximation to its high-fidelity counterpart and can be generated cheaply and abundantly [24]. Many outer-loop applications that require querying the system at many different inputs, including black-box optimization [21], inference [29], and uncertainty propagation [19, 27], can exploit the cross-correlations between low- and high-fidelity data to solve new problems that would otherwise be prohibitively costly to solve using high-fidelity data alone [28, 29].

Motivated by the advent of multifidelity data sources within CSE, in this work we study the reinforcement learning (RL) problem in the presence of multiple environments with different levels of fidelity for a given control task. RL is a popular machine learning paradigm for intelligent sequential decision-making under uncertainty, enabling data-driven control of complex systems with scales ranging from quantum [18] to cosmological [26]. State-of-the-art model-free RL algorithms have indeed demonstrated sheer success for learning complex policies from raw data in single-fidelity environments [25, 22, 31, 32, 12]. This success, however, comes at the cost of requiring a large number of data samples to solve a control task *satisfactorily*.[1] In the presence of multiple environments with different levels of fidelity, new ways arise that could help the agent learn better policies. One way that has been well studied in the context of RL is *transfer learning (TL)*. In TL [35, 8, 39], the agent first uses the low-fidelity environment to learn a policy that is then transferred (directly or indirectly through the transfer of the state-action value function) to the high-fidelity environment as a heuristic to bootstrap learning. Essentially, TL attempts to leverage multifidelity environments to deal with the exploration-exploitation dilemma that is present within RL, and it works under the assumption that the maximum deviation between the optimal low-fidelity state-action value function and the optimal high-fidelity state-action value function is bounded with a threshold that is used by TL for bootsrapping the high-fidelity value function [9]. In our work we explore an uncharted territory and focus on *multifidelity* estimation in RL and its role in improving the learning of the agent. We demonstrate that as long as the low- and high-fidelity state-action value functions for any policy are correlated, significant performance improvements can be reaped by leveraging these cross-correlations without extra effort in managing the exploration-exploitation process.

The main contributions of our work are summarized as follows. First, we study a generic multifidelity setup in which the RL agent can execute a policy in two environments, a low-fidelity environment and a high-fidelity environment. To leverage the cross-correlations between the low- and high-fidelity returns, we propose an unbiased reduced-variance multifidelity estimator for the state-action value function based on the framework of control variates. Second, a multifidelity Monte Carlo (MC) RL algorithm, named `MFMCRL`, is proposed to improve the learning of the RL agent in the high-fidelity environment. For any finite budget of high-fidelity environment interactions, `MFMCRL` leverages low-fidelity data to learn better policies than a standard RL agent that uses only the high-fidelity data. Third, we theoretically analyze the impacts of variance reduction in the estimation of the state-action value function on policy evaluation and policy improvement using probability bounds. Fourth, performance gains of the proposed `MFMCRL` algorithm are empirically assessed through numerical experiments in synthetic multifidelity environments, as well as a neural architecture search (NAS) use case.

## 2 Preliminaries and related work

### 2.1 Reinforcement learning

We consider episodic RL problems where the environment $\Sigma$ is specified by an infinite-horizon Markov decision process (MDP) with discounted returns [5]. Specifically, an infinite-horizon MDP is defined as a tuple $\mathcal{M} = (\mathcal{S}, \mathcal{A}, \mathcal{P}, \beta, \mathcal{R}, \gamma)$, where $\mathcal{S}$ and $\mathcal{A}$ are finite sets of states and actions, respectively; $\mathcal{P} : \mathcal{S} \times \mathcal{A} \times \mathcal{S} \to [0, 1]$ is the environment dynamics; and $\beta : \mathcal{S} \to [0, 1]$ is the initial distribution over the states, that is, $\beta(s) = \Pr(s_0 = s), \forall s \in \mathcal{S}$. The reward function $\mathcal{R}$ is bounded and defined as $\mathcal{R} : \mathcal{S} \times \mathcal{A} \to [R_{\min}, R_{\max}]$, where $R_{\min}$ and $R_{\max}$ are real numbers. $\gamma$ is a discount factor to bound the cumulative rewards and trade off how far- or short-sighted the agent is in its decision making. The environment dynamics, $\mathcal{P}(s'|s, a), \forall s, a, s' \in \mathcal{S} \times \mathcal{A} \times \mathcal{S}$, encode the stationary transition probability from a state $s$ to a state $s'$ given that action $a$ is chosen [7, 16]. In the episodic setting, there exists at least one terminal state $s_T$ such that $\mathcal{P}(s'|s_T, a) = 0, \forall a, s' \neq s_T$ and $\mathcal{P}(s_T|s_T, a) = 1, \forall a$, i.e. $s_T$ is an absorbing state. Furthermore, $\beta(s_T) = 0$ and $\mathcal{R}(s_T, a) = 0, \forall a$. When the RL agent transitions into a terminal state, all subsequent rewards are zero, and simulation is restarted from another state $s \sim \beta$.

The agent's decision-making process is characterized by $\pi(a|s)$, which is a Markov stationary policy that defines a distribution over the actions $a \in \mathcal{A}$ given a state $s \in \mathcal{S}$. In the RL problem, $\mathcal{P}$

---

[1]Poor sample complexity of model-free RL algorithms has long motivated developments in model-based RL, where a predictive model of the environment is learned alongside the policy [14, 30]. Our work is focused on model-free RL.

and $\mathcal{R}$ are not known to the agent, yet the agent can interact with the environment sequentially at discrete time steps, $t = 0, 1, 2, \cdots, T$, by exchanging actions and rewards. Notice that $T$ is a random variable and denotes the time step at which the agent transitions into a terminal state. At each time step $t$, the agent observes the environment's state $s_t = s \in \mathcal{S}$, takes action $a_t = a \sim \pi(a|s) \in \mathcal{A}$, and receives a reward $r_{t+1} = \mathcal{R}(s, a)$. The environment's state then evolves to a new state $s_{t+1} = s' \sim \mathcal{P}(s'|s, a)$. The state-value function of a state $s$ under a policy $\pi$ is defined as the expected long-term discounted returns starting in state $s$ and following policy $\pi$ thereafter,

$V_\pi(s) = \mathbb{E}_{a_t \sim \pi, s_t \sim \mathcal{P}}\left[\sum_{t=0}^\infty \gamma^t \mathcal{R}(s_t, a_t)|s_0 = s\right]$. In addition, the state-action value function of a

state $s$ and action $a$ under a policy $\pi$ is defined as $Q_\pi(s, a) = \mathbb{E}_{a_t \sim \pi, s_t \sim \mathcal{P}}\left[\sum_{t=0}^\infty \gamma^t \mathcal{R}(s_t, a_t)|s_0 = \right.$

$\left. s, a_0 = a\right]$. Notice that $V_\pi(s) = \mathbb{E}_{a \sim \pi}[Q_\pi(s, a)]$. The solution of the RL problem is a policy $\pi^*$ that

maximizes the discounted returns from the initial state distribution $\pi^* = \underset{\pi}{\operatorname{argmax}}\ \mathbb{E}_{s \sim \beta}[V_\pi(s)]$. It is

well known that there exists at least one optimal policy $\pi^*$ such that $V_{\pi^*}(s) = \underset{\pi}{\max}\ V_\pi(s), \forall s \in \mathcal{S}$ and $Q_{\pi^*}(s, a) = \underset{\pi}{\max}\ Q_\pi(s, a), \forall s, a \in \mathcal{S} \times \mathcal{A}$ [2]. Furthermore, a deterministic policy that selects the greedy action with respect to $Q_{\pi^*}(s, a), \forall s \in \mathcal{S}$, is an optimal policy.

## 2.2 Control variates

The method of control variates is a variance reduction technique that leverages the correlation between random variables (r.vs.) to reduce the variance of an estimator [20]. Let $W_1, W_2, \cdots, W_n$ be $n$ independent and identically distributed (i.i.d.) r.vs. such that $\mathbb{E}[W_i] = \mu_W$, and $\mathbb{E}[(W_i - \mu_W)^2] = \sigma_W^2, \forall i \in [n]$. In addition, let $Z_1, Z_2, \cdots, Z_n$ be $n$ i.i.d. r.vs. such that $\mathbb{E}[Z_i] = \mu_z$, and $\mathbb{E}[(Z_i - \mu_z)^2] = \sigma_z^2, \forall i \in [n]$. Suppose that $W_i, Z_i$ are correlated with a correlation coefficient $\rho_{W,Z} = \frac{\operatorname{Cov}[Z_i, W_i]}{\sqrt{\sigma_z^2}\sqrt{\sigma_W^2}}, \forall i \in [n]$, where $\operatorname{Cov}[Z_i, W_i] = \mathbb{E}[Z_i W_i] - \mathbb{E}[Z_i]\mathbb{E}[W_i]$ is the covariance between $Z_i$ and $W_i$. Furthermore, suppose that $W_i, Z_j$ are independent and thus uncorrelated $\forall i \neq j$. Using the Cauchy—Schwartz inequality, one can show that $|\rho_{W,Z}| \leq 1$.

To estimate $\mu_W$, we first consider the sample mean estimator, $\hat{\theta}_1 = \frac{1}{n}\sum_{i=1}^n W_i$. $\hat{\theta}_1$ is an unbiased estimator of $\mu_W$, in other words, $\mathbb{E}[\hat{\theta}_1] = \frac{1}{n}\sum_{i=1}^n \mathbb{E}[W_i] = \mu_W$, and has a variance $\operatorname{Var}[\hat{\theta}_1] = \frac{\sigma_W^2}{n}$. Next, we consider the control-variate-based estimator,

$$\hat{\theta}_2 = \frac{1}{n}\sum_{i=1}^n W_i + \alpha(Z_i - \mu_z). \tag{1}$$

$\hat{\theta}_2$ is also an unbiased estimator of $\mu_W$, i.e., $\mathbb{E}[\hat{\theta}_2] = \mu_W$, yet it has a variance $\operatorname{Var}[\hat{\theta}_2] = \frac{1}{n}\operatorname{Var}[W_i + \alpha(Z_i - \mu_z)] = \frac{1}{n}\left(\operatorname{Var}[W_i] + \alpha^2\operatorname{Var}[Z_i] + 2\alpha\operatorname{Cov}[Z_i, W_i]\right)$. The variance of $\hat{\theta}_2$ can be controlled and minimized by setting $\alpha$ to the minima of $\operatorname{Var}[W_i] + \alpha^2\operatorname{Var}[Z_i] + 2\alpha\operatorname{Cov}[Z_i, W_i]$, which is attained at $\alpha^* = -\frac{\operatorname{Cov}[Z_i, W_i]}{\sigma_z^2} = -\rho_{z,W}\frac{\sigma_W}{\sigma_z}$. Hence, by introducing $\alpha(Z_i - \mu_z)$ as a control variate, the variance of $\hat{\theta}_2$ is reduced,

$$\operatorname{Var}[\hat{\theta}_2] = (1 - \rho_{z,W}^2)\operatorname{Var}[\hat{\theta}_1]. \tag{2}$$

Because $\hat{\theta}_2$ is an unbiased estimator, $\hat{\theta}_2$ has a lower mean squared error (MSE) by the bias-variance decomposition theorem of the MSE. Applications of the method of control variates extend beyond variance reduction. For example, the concept of control variates is used in [27] to design a fusion framework to combine an arbitrary number of surrogate models optimally.

## 2.3 Related work

In [1], a policy search algorithm is proposed that leverages a crude approximate model $\hat{\mathcal{P}}$ of the true MDP to quickly learn to perform well on real systems. The proposed algorithm, however, is limited to the case where $\mathcal{P}$ is deterministic, and it assumes that model derivatives are good approximations of the true derivatives such that policy gradients can be computed by using the approximate model.

In transfer learning (TL) [36, 23], value, model, or policy parameters are transferred in one direction as a heuristic initialization to bootstrap learning in the high-fidelity environment, with no option for backtracking. The option for the agent to backtrack and to choose which environment to use is studied in the multifidelity RL (MFRL) work of [9]. That algorithm is extended in [33] by integrating function approximation using Gaussian processes [38]. As in TL, both [9] and [33] use the value function from a lower-fidelity environment as a heuristic to bootstrap learning and *guide exploration* in the high-fidelity environment. From an optimization viewpoint, this approach is reasonable only if the lower-fidelity value function lies in the vicinity of the optimal high-fidelity value function, a situation that cannot be guaranteed or known a priori in general. Hence, in [9, 33], it is assumed that the optimal state-action value function in the low- and high-fidelity environments differ by no more than a small parameter $\beta$ at every state-action pair, and they require the knowledge of $\beta$ a priori to manage exploration-exploitation across multifidelity environments. By contrast, we require only that the low- and high-fidelity returns are correlated in our work, and the correlation need not be known a priori. The cross-correlation between the low- and high-fidelity returns is used for reducing the variance in the *estimation* of the high-fidelity state-action value function, and hence our approach is complementary to existing TL techniques that use multifidelity environments for guided exploration [9, 33]. We show that as long as the low- and high-fidelity state-action value function of a policy are correlated, the agent can benefit from the cheap and abundantly available low-fidelity data to improve its performance, without altering the exploration process.

# 3  Multifidelity estimation in RL

## 3.1  Problem setup

We consider a multifidelity setup in which the RL agent has access to two environments, $\Sigma^{\text{lo}}$ and $\Sigma^{\text{hi}}$, modeled by the two MDPs $\mathcal{M}^{\text{lo}} = (\mathcal{S}^{\text{lo}}, \mathcal{A}, \mathcal{P}^{\text{lo}}, \beta^{\text{lo}}, \mathcal{R}^{\text{lo}}, \gamma)$, and $\mathcal{M}^{\text{hi}} = (\mathcal{S}^{\text{hi}}, \mathcal{A}, \mathcal{P}^{\text{hi}}, \beta^{\text{hi}}, \mathcal{R}^{\text{hi}}, \gamma)$, respectively, as shown in Figure 1. $\Sigma^{\text{lo}}$ is a low-fidelity environment in which the low-fidelity reward function $\mathcal{R}^{\text{lo}} : \mathcal{S} \times \mathcal{A} \to [R_{\min}^{\text{lo}}, R_{\max}^{\text{lo}}]$ and the low-fidelity dynamics $\mathcal{P}^{\text{lo}}$ are cheap[2] to evaluate/simulate, yet they are potentially inaccurate. On the other hand, $\Sigma^{\text{hi}}$ is a high-fidelity environment in which the high-fidelity reward function $\mathcal{R}^{\text{hi}} : \mathcal{S} \times \mathcal{A} \to [R_{\min}^{\text{hi}}, R_{\max}^{\text{hi}}]$ and the high-fidelity dynamics $\mathcal{P}^{\text{hi}}$ describe the real-world system with the highest accuracy, yet they are expensive to evaluate/simulate [11]. We stress that $(\mathcal{P}^{\text{hi}}, \beta^{\text{hi}}, \mathcal{R}^{\text{hi}})$ and $(\mathcal{P}^{\text{lo}}, \beta^{\text{lo}}, \mathcal{R}^{\text{lo}})$ are **unknown** to the agent, and interaction with the two environments is only through the exchange of states, actions, next states and rewards, which is the typical case in RL.

The action space $\mathcal{A}$ is the same in both environments, yet the state space may differ. It is assumed that the low-fidelity state space is a subset of the high-fidelity state space, $\mathcal{S}^{\text{lo}} \subseteq \mathcal{S}^{\text{hi}}$, in other words, the states available in the low-fidelity environment are a subset of those available at the high-fidelity environment, and it is assumed that there exists a known mapping[3] $\mathcal{T} : \mathcal{S}^{\text{hi}} \to \mathcal{S}^{\text{lo}}$ as in previous works [36, 9]. High-fidelity environments usually capture more state information than do low- fidelity environments so $\mathcal{T}$ can be a many-to-one map. Access to the high-fidelity simulator $\Sigma^{\text{hi}}$ is restricted to full episodes $\tau^{\text{hi}} = (s_0^{\text{hi}}, a_0, r_1^{\text{hi}}, s_1^{\text{hi}}, a_1, r_2^{\text{hi}}, s_2^{\text{hi}}, \cdots, s_T^{\text{hi}})$. On the other hand, $\Sigma^{\text{lo}}$ is generative, and simulation can be started by the agent at any state-action pair [15, 17]. Using $\mathcal{T}$ and $\Sigma^{\text{lo}}$, the agent can map a $\tau^{\text{hi}}$ to $\tau^{\text{lo}} = (\mathcal{T}(s_0^{\text{hi}}), a_0, r_1^{\text{lo}}, \mathcal{T}(s_1^{\text{hi}}), a_1, r_2^{\text{lo}}, \mathcal{T}(s_2^{\text{hi}}), \cdots, \mathcal{T}(s_T^{\text{hi}}))$, and it is assumed that $\Pr(\tau^{\text{lo}}) > 0$ under $\mathcal{P}^{\text{lo}}$ and $\beta^{\text{lo}}$. It is also assumed that $\mathcal{R}^{\text{lo}}(\mathcal{T}(s^{\text{hi}}), a)$ and $\mathcal{R}^{\text{hi}}(s^{\text{hi}}, a)$ are correlated.

Based on this setup, a correlation exits between the low- and high- fidelity trajectories that can be beneficial for policy learning. In this work we study how to leverage the cheaply accessible low-fidelity trajectories from $\Sigma^{\text{lo}}$, to learn an optimal $\pi^*$ that maximizes $\mathbb{E}_{s \sim \beta^{\text{hi}}} \left[ \mathbb{E}_{a_t \sim \pi, s_t \sim \mathcal{P}^{\text{hi}}} \left[ \sum_{t=0}^{\infty} \gamma^t \mathcal{R}^{\text{hi}}(s_t^{\text{hi}}, a_t) | s_0^{\text{hi}} = s \right] \right]$; in other words, to learn $\pi^*$ that is optimal with respect to the high-fidelity environment $\Sigma^{\text{hi}}$.

---

[2]Sampling cost is application dependent. It is up to the practitioner to assign cost and determine low- and high-fidelity sampling budgets.

[3]$\mathcal{T}$ is problem-specific. For instance, if $\mathcal{S}^{\text{hi}}$ represents a fine grid and $\mathcal{S}^{\text{lo}}$ represents a coarse grid, then $\mathcal{T}$ will map $s^{\text{hi}}$ to the closest $s^{\text{lo}}$ based on a chosen distance metric.

## 3.2 Multifidelity Monte Carlo RL

The Monte Carlo method to solve the RL prob-
lem is based on the idea of averaging sample
returns. In the MC method, experience is di-
vided into episodes. At the end of an episode,
state-action values are estimated, and the policy
is updated. For ease of exposition, we consider a
specific state-action pair $(s^{\mathrm{hi}}, a)$ in what follows
and suppress the dependence on $(s^{\mathrm{hi}}, a)$ from
the notation to avoid clutter. Consider a sam-
ple trajectory $\tau^{\mathrm{hi}}$ that results from the agent's
interaction with the high-fidelity environment
starting at $(s_0^{\mathrm{hi}} = s^{\mathrm{hi}}, a_0 = a)$ and following
$\pi$, that is, $\tau^{\mathrm{hi}} : s_0^{\mathrm{hi}}, a_0, r_1^{\mathrm{hi}}, s_1^{\mathrm{hi}}, a_1, r_2^{\mathrm{hi}}, \cdots, s_T^{\mathrm{hi}}$.

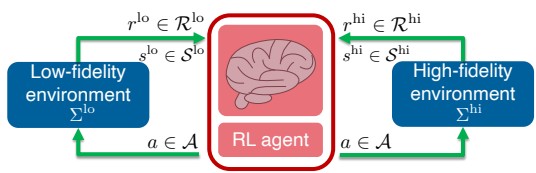

Figure 1: RL with low- and high-fidelity environ-
ments. $\Sigma^{\mathrm{lo}}$ is cheap to evaluate but is potentially
inaccurate. $\Sigma^{\mathrm{hi}}$ represents the real world with the
highest accuracy, yet it is expensive to evaluate.
The RL agent leverages the correlations between
the low- and high-fidelity data to learn $\pi_{\mathrm{hi}}^*$.

Note that $r_{t+1}^{\mathrm{hi}} = \mathcal{R}^{\mathrm{hi}}(s_t^{\mathrm{hi}}, a_t)$. Let $\mathcal{G}^{\mathrm{hi}}$ denote the corresponding long-term discounted return,
$\mathcal{G}^{\mathrm{hi}} = \sum_{t=0}^{\infty} \gamma^t r_{t+1}^{\mathrm{hi}}$. The high-fidelity state-action value of the pair $(s, a)$ when the agent follows $\pi$
is

$$Q_\pi^{\mathrm{hi}}(s^{\mathrm{hi}}, a) = \mathbb{E}_{\tau^{\mathrm{hi}}}\big[\mathcal{G}^{\mathrm{hi}}|s_0^{\mathrm{hi}} = s^{\mathrm{hi}}, a_0 = a\big]. \tag{3}$$

Notice that $Q_\pi^{\mathrm{hi}}(s^{\mathrm{hi}}, a)$ is the expectation of an r.v. $\mathcal{G}^{\mathrm{hi}}$ with respect to the random trajectory $\tau^{\mathrm{hi}}$. $\mathcal{G}^{\mathrm{hi}}$
is a bounded r.v. with support on the interval $\big[\frac{R_{\min}^{\mathrm{hi}}}{1-\gamma}, \frac{R_{\max}^{\mathrm{hi}}}{1-\gamma}\big]$ and has a finite variance given by

$$\sigma_{\mathrm{hi}}^2(s^{\mathrm{hi}}, a) = \mathbb{E}_{\tau^{\mathrm{hi}}}\Big[\big(\mathcal{G}^{\mathrm{hi}} - Q_\pi^{\mathrm{hi}}(s^{\mathrm{hi}}, a)\big)^2|s_0 = s^{\mathrm{hi}}, a_0 = a\Big]. \tag{4}$$

By interacting with the environment, the agent can sample only a finite number of trajectories, $n$.
Let $\tau_1^{\mathrm{hi}}, \tau_2^{\mathrm{hi}}, \cdots, \tau_n^{\mathrm{hi}}$ be the $n$ sampled trajectories that starts at the pair $(s^{\mathrm{hi}}, a)$. Furthermore, let
$\mathcal{G}_1^{\mathrm{hi}}, \mathcal{G}_2^{\mathrm{hi}}, \cdots, \mathcal{G}_n^{\mathrm{hi}}$ be i.i.d. r.vs. that correspond to the long-term discounted returns of the sampled
trajectories, $\tau_1^{\mathrm{hi}}, \tau_2^{\mathrm{hi}}, \cdots, \tau_n^{\mathrm{hi}}$, respectively. Notice that $\mathbb{E}_{\tau^{\mathrm{hi}}}[\mathcal{G}_1^{\mathrm{hi}}] = \mathbb{E}_{\tau^{\mathrm{hi}}}[\mathcal{G}_2^{\mathrm{hi}}] = \cdots = \mathbb{E}_{\tau^{\mathrm{hi}}}[\mathcal{G}_n^{\mathrm{hi}}] =$
$Q_\pi^{\mathrm{hi}}(s, a)$. The first-visit MC sample average is

$$\hat{Q}_{\pi,n}^{\mathrm{hi}}(s^{\mathrm{hi}}, a) = \frac{1}{n}\sum_{i=1}^n \mathcal{G}_i^{\mathrm{hi}}. \tag{5}$$

By the weak law of large numbers, $\lim_{n\to\infty} \Pr\big(|\hat{Q}_{\pi,n}^{\mathrm{hi}}(s^{\mathrm{hi}}, a) - Q_\pi^{\mathrm{hi}}(s^{\mathrm{hi}}, a)| > \xi\big) = 0$, for any positive
number $\xi$. In addition, the variance of this unbiased sample average estimator is

$$\mathrm{Var}\Big[\hat{Q}_{\pi,n}^{\mathrm{hi}}(s^{\mathrm{hi}}, a)\Big] = \frac{\sigma_{\mathrm{hi}}^2(s^{\mathrm{hi}}, a)}{n}. \tag{6}$$

Using the low-fidelity generative environment and the method of control variates, we design an
unbiased estimator for the expected long-term discounted returns that has a smaller variance than
(6). Let $\tau_i^{\mathrm{lo}}$ be the $i$th low-fidelity trajectory that is obtained from $\tau_i^{\mathrm{hi}}$ by using $\mathcal{T}$ and the generative
low-fidelity environment to evaluate $r_{t+1}^{\mathrm{low}} = \mathcal{R}^{\mathrm{lo}}(\mathcal{T}(s_t^{\mathrm{hi}}), a_t)$. Let $\mathcal{G}_i^{\mathrm{lo}}$ be the r.v. which corresponds to
the long-term discounted return of $\tau_i^{\mathrm{lo}}$. Notice that $\mathcal{G}_i^{\mathrm{hi}}$ and $\mathcal{G}_i^{\mathrm{lo}}$ are correlated r.vs. in this multifidelity
setup. Based on those low-fidelity trajectories, the low-fidelity first-visit MC sample average is
$\hat{Q}_{\pi,n}^{\mathrm{lo}}(\mathcal{T}(s^{\mathrm{hi}}), a) = \frac{1}{n}\sum_{i=1}^n \mathcal{G}_i^{\mathrm{lo}}$ and has a variance of $\mathrm{Var}\Big[\hat{Q}_{\pi,n}^{\mathrm{lo}}(\mathcal{T}(s^{\mathrm{hi}}), a)\Big] = \frac{\sigma_{\mathrm{lo}}^2(\mathcal{T}(s^{\mathrm{hi}}), a)}{n}$, where
$\sigma_{\mathrm{lo}}^2(\mathcal{T}(s^{\mathrm{hi}}), a) = \mathbb{E}_{\tau^{\mathrm{lo}}}\Big[\big(\mathcal{G}^{\mathrm{lo}} - Q_\pi^{\mathrm{lo}}(\mathcal{T}(s^{\mathrm{hi}}), a)\big)^2|s_0 = \mathcal{T}(s^{\mathrm{hi}}), a_0 = a\Big]$ and $Q_\pi^{\mathrm{lo}}(\mathcal{T}(s^{\mathrm{hi}}), a)$ is the true
population mean.

Using the method of control variates presented in Subsection 2.2, we propose the following multifi-
delity MC estimator:

$$\hat{Q}_{\pi,n}^{\mathrm{MFMC}}(s^{\mathrm{hi}}, a) = \hat{Q}_{\pi,n}^{\mathrm{hi}}(s^{\mathrm{hi}}, a) + \alpha_{s,a}^*\Big(Q_\pi^{\mathrm{lo}}(\mathcal{T}(s^{\mathrm{hi}}), a) - \hat{Q}_{\pi,n}^{\mathrm{lo}}(\mathcal{T}(s^{\mathrm{hi}}), a)\Big), \tag{7}$$

where

$$\alpha_{s,a}^* = \frac{\mathrm{Cov}\big[\hat{Q}_{\pi,n}^{\mathrm{hi}}(s^{\mathrm{hi}}, a), \hat{Q}_{\pi,n}^{\mathrm{lo}}(\mathcal{T}(s^{\mathrm{hi}}), a)\big]}{\mathrm{Var}\big[\hat{Q}_{\pi,n}^{\mathrm{lo}}(\mathcal{T}(s^{\mathrm{hi}}), a)\big]}. \tag{8}$$

Notice that the estimator in (7) is unbiased and has a variance of

$$\text{Var}\Big[\hat{Q}_{\pi,n}^{\text{MFMC}}(s^{\text{hi}}, a)\Big] = \big(1 - \rho_{s,a}^2\big)\text{Var}\Big[\hat{Q}_{\pi,n}^{\text{hi}}(s^{\text{hi}}, a)\Big], \tag{9}$$

where $\rho_{s,a}$ is the correlation coefficient between the low-fidelity and high-fidelity long-term discounted returns:

$$\rho_{s,a} = \frac{\text{Cov}\big[\hat{Q}_{\pi,n}^{\text{hi}}(s^{\text{hi}}, a), \hat{Q}_{\pi,n}^{\text{lo}}(\mathcal{T}(s^{\text{hi}}), a)\big]}{\sqrt{\text{Var}\Big[\hat{Q}_{\pi,n}^{\text{hi}}(s^{\text{hi}}, a)\Big]\text{Var}\Big[\hat{Q}_{\pi,n}^{\text{lo}}(\mathcal{T}(s^{\text{hi}}), a)\Big]}}. \tag{10}$$

Therefore, the variance in estimating the value of a state-action pair under a policy $\pi$ can be reduced by a factor of $\big(1 - \rho_{s,a}^2\big)$ when the low-fidelity data is exploited, although the budget of high-fidelity samples remains the same. Notice that

$$\text{Cov}\big[\hat{Q}_{\pi,n}^{\text{hi}}(s^{\text{hi}}, a), \hat{Q}_{\pi,n}^{\text{lo}}(\mathcal{T}(s^{\text{hi}}), a)\big] = \text{Cov}\Big[\frac{1}{n}\sum_{i=1}^{n}\mathcal{G}_i^{\text{hi}}, \frac{1}{n}\sum_{i=1}^{n}\mathcal{G}_i^{\text{lo}}\Big] = \frac{1}{n}\text{Cov}\big[\mathcal{G}_i^{\text{hi}}, \mathcal{G}_i^{\text{lo}}\big], \tag{11}$$

because $\mathcal{G}_i^{\text{hi}}, \mathcal{G}_j^{\text{lo}}$ are independent r.vs. $\forall i \neq j$. Hence, $\text{Cov}\big[\hat{Q}_{\pi,n}^{\text{hi}}(s^{\text{hi}}, a), \hat{Q}_{\pi,n}^{\text{lo}}(\mathcal{T}(s^{\text{hi}}), a)\big]$, $\text{Var}\Big[\hat{Q}_{\pi,n}^{\text{hi}}(s^{\text{hi}}, a)\Big]$, and $\text{Var}\Big[\hat{Q}_{\pi,n}^{\text{lo}}(\mathcal{T}(s^{\text{hi}}), a)\Big]$ can all be estimated in practice based on the return data samples using the standard unbiased estimators for the variance and covariance.

The reduced-variance estimator of (7) can be used to design a multifidelity Monte Carlo RL algorithm as shown in Algorithm 1 in Appendix A. This algorithm is based on the on-policy first-visit MC control algorithm with $\epsilon$-soft policies [34] but uses the multifidelity estimator (7). Algorithm 1 is based on the idea of generalized policy iteration. In the policy evaluation step (lines 11–18), the state-action value function is made consistent with the current policy by updating the estimated long-term discounted returns of a state-action pair $(s_t, a_t)$ using the control-variate-based estimator (7) (line 18). This update requires the estimation of the correlation between the low- and high-fidelity returns, which is done in lines 13–17. Next, in the policy improvement step (lines 19–20), the policy is made $\epsilon$-greedy with respect to the current state-action value function. In each episode, the agent needs to evaluate the policy in the low-fidelity environment to obtain $Q_\pi^{\text{lo}}$. This can be done in practice by collecting a large number of $m$ return samples from the cheap low-fidelity environment and setting $Q_\pi^{\text{lo}}(\mathcal{T}(s^{\text{hi}}), a) \approx \hat{Q}_{\pi,m+n}^{\text{lo}}(\mathcal{T}(s^{\text{hi}}), a)$. The convergence of Algorithm 1 to the optimal $\epsilon$-greedy policy, $\pi_{\epsilon-\text{opt}}^*$, along with its corresponding $\hat{Q}_*^{\text{MFMC}}$, is guaranteed under the same conditions that guarantee convergence for the on-policy first-visit MC control algorithm with $\epsilon$-soft policies [34]. In the following subsection, we theoretically analyze the impacts of variance reduction on policy evaluation and policy improvement.

## 3.3 Theoretical analysis

In this subsection we analyze the impacts of variance reduction on policy evaluation error and policy improvement by introducing two main theorems. Intermediate lemmas along with all the proofs can be found in Appendix B.

### 3.3.1 Policy evaluation

In policy evaluation, the task is to estimate the state-action value function of a given policy $\pi$. Trajectory samples are first generated by interacting with the environment using $\pi$, and the state-action value function is then estimated using either the single high-fidelity estimator (5) or the proposed multifidelity estimator (7). To analyze the impacts of variance reduction on policy evaluation error, we first derive a a Bernstein-type concentration inequality [6] that relates the deviation between the sample average and the true mean to the sample size $n$, estimation accuracy parameters $\delta, \xi$, and the variance of a r.v. as follows.

**Lemma 1** *Let $X_1, X_2, \cdots, X_n$ be i.i.d. r.vs. with mean $\mathbb{E}[X_i] = \mu_X$ and variance $\mathbb{E}[(X_i - \mu_X)^2] = \sigma_X^2$, $\forall i \in [n]$. Furthermore, suppose that $X_i, \forall i$, are bounded almost surely with a parameter $b$, namely, $Pr(|X_i - \mu_X| \leq b) = 1, \forall i$. Then*

$$Pr\left(\left|\frac{1}{n}\sum_{i=1}^{n}X_i - \mu_X\right| \geq \xi\right) \leq 2exp\left(\frac{-n\xi^2}{4\sigma_X^2}\right) \tag{12}$$

253  *for* $0 \leq \xi \leq \sigma_X^2/b$.

254  Next, the concentration bound of Lemma 1 is used to derive the minimum sample size that is required
255  to ensure that the sample average deviates by no more than $\xi$ from the true mean with high probability
256  for both the high-fidelity estimator (5) and the multifidelity estimator (7).

**Theorem 1** *To guarantee that*

257

258  *1.* $Pr\Big(|\hat{Q}_{\pi,n}^{hi}(s^{hi},a) - Q_\pi^{hi}(s^{hi},a)| \leq \xi\Big) \geq 1 - \delta$, *then* $n \geq \frac{4\sigma_{hi}^2(s^{hi},a)}{\xi^2}log(\frac{2}{\delta})$.

259  *2.* $Pr\Big(|\hat{Q}_{\pi,n}^{MFMC}(s,a) - Q_\pi^{hi}(s^{hi},a)| \leq \xi\Big) \geq 1 - \delta$, *then* $n \geq \frac{4(1-\rho_{s,a}^2)\sigma_{hi}^2(s^{hi},a)}{\xi^2}log(\frac{2}{\delta})$.

260  The result of Theorem 1 highlights the benefit of using our proposed multifidelity estimator (7) for
261  policy evaluation as opposed to using the single high-fidelity estimator of (5). By leveraging the
262  correlation between low- and high-fidelity returns $\rho_{s,a}$, the variance of the multifidelity estimator
263  is reduced by a factor of $(1 - \rho_{s,a}^2)$, which makes it possible to achieve a low estimation error at a
264  reduced number of high-fidelity samples.

### 3.3.2 Policy improvement

265

266  In policy improvement, a new policy $\pi'$ is constructed by deterministically choosing the greedy
267  action with respect to the state-action value function of the original policy $\pi$, $Q_\pi^{hi}(s,a)$, at every state,
268  that is, $\pi'(s) \doteq \underset{a \in \mathcal{A}}{\text{argmax}} \, Q_\pi^{hi}(s,a), \forall s \in \mathcal{S}$. By the policy improvement theorem, $\pi'$ is as good as or
269  better than $\pi$ under the assumption that $Q_\pi^{hi}(s,a), \forall s,a$ is computed exactly. In practice, the MDP is
270  unknown, and the state-action value function is estimated based on a finite number of trajectories.
271  Moreover, those trajectories are generated by following an exploratory policy, such as an $\epsilon$-soft
272  policy. Because we are interested in studying how different estimators impact policy improvement,
273  we consider a target state $s^{hi} \in \mathcal{S}^{hi}$ and assume that we have $n$ trajectories for each action $a \in \mathcal{A}$ at
274  this target state. This assumption basically ensures that all actions at the target state $s^{hi}$ have been
275  explored equally well and enables us to make fair comparisons about estimator performance.

276  Without loss of generality, suppose that $Q_\pi^{hi}(s^{hi},a_1) \geq Q_\pi^{hi}(s^{hi},a_2) \geq \cdots Q_\pi^{hi}(s^{hi},a_{|\mathcal{A}|})$. Let $\Delta_i =$
277  $Q_\pi^{hi}(s^{hi},a_1) - Q_\pi^{hi}(s^{hi},a_i), \forall i \neq 1$. We analyze the probability that $a_1$, which is the greedy action
278  given the true $Q_\pi^{hi}(s^{hi},a)$, is the greedy action with respect to the single- and multifidelity estimators
279  in our next theorem.

280  **Theorem 2** *Suppose that the number of trajectories from a state-action pair at a target state $s^{hi} \in \mathcal{S}^{hi}$*
281  *is the same for all actions $a \in \mathcal{A}$ and that $a_1$ is the greedy action with respect to the true $Q_\pi^{hi}(s^{hi},a)$.*
282  *Furthermore, suppose that $\mathcal{P}^{hi}(s^{hi}|s^{hi'},a) \geq \beta(s^{hi}), \forall s^{hi} \in \mathcal{S}^{hi}$. Then*

283  *1.* $Pr\Big(a_1 = \underset{a \in \mathcal{A}}{\text{argmax}} \, \hat{Q}_{\pi,n}^{hi}(s^{hi},a)\Big) \geq \prod_{i=2}^{|\mathcal{A}|} \frac{\Delta_i^2}{\Delta_i^2 + Var[\hat{Q}_{\pi,n}^{hi}(s^{hi},a_1)] + Var[\hat{Q}_{\pi,n}^{hi}(s^{hi},a_i)]}$.

284  *2.* $Pr\Big(a_1 = \underset{a \in \mathcal{A}}{\text{argmax}} \, \hat{Q}_{\pi,n}^{MFMC}(s^{hi},a)\Big) \geq \prod_{i=2}^{|\mathcal{A}|} \frac{\Delta_i^2}{\Delta_i^2 + (1-\rho_{s,a_1}^2)Var[\hat{Q}_{\pi,n}^{hi}(s^{hi},a_i)] + (1-\rho_{s,a_i}^2)Var[\hat{Q}_{\pi,n}^{hi}(s^{hi},a_i)]}$.

285  Notice that when $|\rho_{s,a_2}| \to 1$, the lower bound in the result of Theorem 2 approaches 1, which
286  means that the correct greedy action $a_1$ can be selected with certainty when the reduced-variance
287  multifidelity estimator (7) is adopted. Combining the results of Theorems 1 and 2, the proposed
288  MFMCRL algorithm is expected to outperform its single high-fidelity Monte Carlo counterpart in terms
289  of learning a better policy under a given budget of high-fidelity environment interactions.

## 4 Numerical experiments

290

291  In this section we empirically evaluate the performance of the proposed MFMCRL algorithm on
292  synthetic MDP problems and on a NAS use case. Our codes and all experimental details can be found
293  in Appendix C.

## 4.1 Synthetic MDPs

We synthesize multifidelity random MDP problems with state space cardinality $|\mathcal{S}|$ and action space cardinality $|\mathcal{A}|$. The high-fidelity transition and reward functions, $\mathcal{P}^{\text{hi}}$ and $\mathcal{R}^{\text{hi}}$, respectively, are first generated based on a random process as detailed in Appendix C.2. Next, for a given $\mathcal{P}^{\text{hi}}$ and $\mathcal{R}^{\text{hi}}$, the corresponding $\mathcal{P}^{\text{low}}$ and $\mathcal{R}^{\text{low}}$ are generated by injecting Gaussian noise to meet a desired signal-to-noise ratio. Specifically, we generate a random matrix $\mathcal{P}_N$ of size $|\mathcal{S}| \times |\mathcal{A}| \times |\mathcal{S}|$ from a normally distributed r.v. with mean 0 and variance $\sigma_\mathcal{P}^2$, and set $\mathcal{P}^{\text{low}} = \mathcal{P}^{\text{hi}} + \mathcal{P}_N$. $\mathcal{P}^{\text{low}}$ is then appropriately normalized so that $\sum_{s^{\text{lo}'} \in \mathcal{S}} \mathcal{P}^{\text{lo}}(s^{\text{lo}'}|s^{\text{lo}}, a) = 1$. Similarly, we generate a random matrix $\mathcal{R}_N$ of size $|\mathcal{S}| \times |\mathcal{A}|$ from a normally distributed r.v. with mean 0 and variance $\sigma_\mathcal{R}^2$ and set $\mathcal{R}^{\text{low}} = \mathcal{R}^{\text{hi}} + \mathcal{R}_N$. $\mathcal{P}^{\text{hi}}$ and $\mathcal{R}^{\text{hi}}$ are then encapsulated within a gym-like environment with which the agent can interact by exchanging sample tuples of the form $(s^{\text{hi}}, a, r^{\text{hi}}, s^{\text{hi}'})$. Similarly, $\mathcal{P}^{\text{lo}}$ and $\mathcal{R}^{\text{lo}}$ are encapsulated within a gym-like environment to form the low-fidelity environment. In this experiment, both low- and high-fidelity environments share the same state-action space—that is, $\mathcal{T}$ is an identity transformation—yet the transition and reward functions of the low-fidelity environment are different since they are corrupted with noise. Notice that even if the agent could draw an infinite number of samples from $\mathcal{P}^{\text{lo}}$ and $\mathcal{R}^{\text{lo}}$, it would not be able to recover $\mathcal{P}^{\text{hi}}$ and $\mathcal{R}^{\text{hi}}$ since $\mathcal{P}^{\text{lo}}$ and $\mathcal{R}^{\text{lo}}$ underneath the low-fidelity environment themselves are corrupted. This situation mimics what happens in practice when we attempt to learn $\mathcal{P}^{\text{lo}}$ and $\mathcal{R}^{\text{lo}}$ based on real data and build an RL environment off those learned functions to train the agent.

After constructing the multifidelity environments, we train an RL agent using the proposed MFMCRL algorithm over 10K high-fidelity episodes, where a training episode is defined to be a trajectory that ends at a terminal state. The MFMCRL agent interacts with the low-fidelity environment as shown in Algorithm 1, to generate reduced-variance estimates of the state-action value function. As a baseline for comparison, we train another RL agent (MCRL) using the standard the first-visit MC control algorithm over 10K high-fidelity episodes [34]. We set $\gamma$ and $\epsilon$ to 0.99 and 0.1, respectively. Every 50 training episodes, the greedy policy w.r.t to the estimated $Q$ function is used to test the performance of the agent on 200 test episodes. We repeat the whole experiment with 36 different random seeds (to fully leverage our 36 core machine) and report the mean and standard deviation (across different seeds) of the test episode rewards in Figure 2(a). One can observe that for a given budget of high-fidelity episodes, the proposed MFMCRL algorithm outperforms MCRL in terms of policy performance, with performance improving as the RL agent collects more low-fidelity samples ($\#\tau^{\text{lo}}$ refers to the number of low-fidelity trajectories started from a state-action pair). In Figure 2(b), we vary the SNR of the low-fidelity environment and observe that performance improves as SNR increases. This is expected because the low- and high-fidelity environments are better-correlated at higher SNRs. Notice that when the SNR of the low-fidelity environment is -10 dB, there is no benefit from doing multifidelity RL. The reason is that the low- and high-fidelity environments are too weakly correlated to benefit from multifidelity estimation. In fact, for this case $\mathbb{E}_{s,a,s'}[|\mathcal{P}^{\text{hi}} - \mathcal{P}^{\text{lo}}|] = 0.275 \pm 0.33$, and $\mathbb{E}_{s,a}[|\mathcal{R}^{\text{hi}} - \mathcal{R}^{\text{lo}}|] = 1.029 \pm 0.024$, compared with the other extreme case (SNR +3dB) for which $\mathbb{E}_{s,a,s'}[|\mathcal{P}^{\text{hi}} - \mathcal{P}^{\text{lo}}|] = 0.009 \pm 0.0002$, and $\mathbb{E}_{s,a}[|\mathcal{R}^{\text{hi}} - \mathcal{R}^{\text{lo}}|] = 0.230 \pm 0.006$. This is also evident in Figure 2(c), where we show the mean variance reduction factor $\text{Var}[\hat{Q}^{\text{MFMC}}]/\text{Var}[\hat{Q}^{\text{hi}}]$ estimated based off the last 1K training episodes. When the low-fidelity environment is less noisy (higher SNR), more variance reduction can be attained.

## 4.2 NAS

In NAS, the task is to discover high-performing neural architectures with respect to a given training dataset over a predefined search space. While many earlier works attempted to design RL-based NAS algorithms, [3, 40, 13], it has since become clear that the sample complexity of RL is too high to be competitive with state-of-the-art NAS methods [4, 37]. In this experiment we study how multifidelity RL can improve learning in NAS over standard RL, which could serve to catalyze future work in this direction to make RL more competitive in NAS.

For this experiment we use the tabular dataset of NAS-Bench-201 [10] to construct multifidelity RL environments as detailed in Appendix C.3. In summary, the RL agent sequentially configures the nodes of an architecture (inducing an MDP), after which the architecture is trained on the training dataset for $L$ epochs, and the validation accuracy on a held-out validation data set is provided to the agent as a reward. By maximizing the total rewards, high-performing architectures can be discovered. NAS-Bench-201 reports the validation accuracy curves for all the architectures in the search space

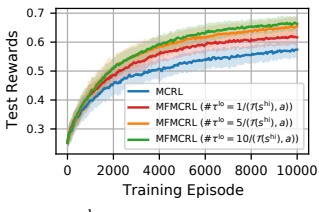
(a) $\Sigma^{\text{lo}}$ has an SNR of -3dB

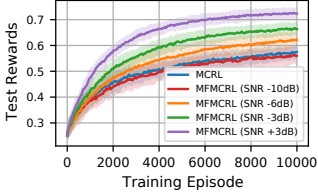
(b) $\#\tau^{\text{lo}} = 10/(\mathcal{T}(s^{\text{hi}}), a))$

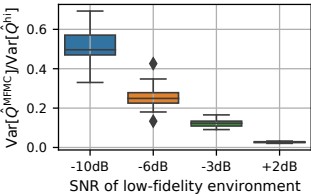
(c) Variance reduction factor

Figure 2: Mean and standard deviation of test episode rewards for the proposed MFMCRL during training: (a) test episode rewards improve with increasing number of low-fidelity samples ($\#\tau^{\text{lo}}$); (b) test episode rewards improve with less noisy low-fidelity environments; (c) variance reduction factor improves when low- and high-fidelity environments are more correlated. These results are based on a random MDP with $|\mathcal{S}| = 200, |\mathcal{A}| = 8$.

as a function of the number of training epochs and for three image data sets. We construct two multifidelity scenarios as follows. In both scenarios, the validation accuracy of an architecture at the end of training (i.e. at $L = 200$ epochs) is used as a high-fidelity reward in the high-fidelity environment. For the low-fidelity environment, we have two cases: (i) low-fidelity environment is identical to the high-fidelity environment except for the reward function, which is now the validation accuracy at the $L = 10$th training epoch, and (ii) low-fidelity environment is defined for a smaller search space and the reward function is the validation accuracy of an architecture at the $L = 10$th training epoch. Note that in case (ii) the state space and dynamics differ between the low- and high-fidelity envi-

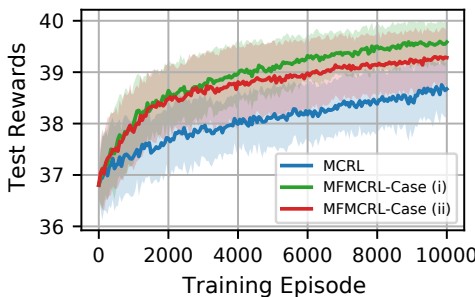

Figure 3: Mean and standard deviation of test episode rewards for the proposed MFMCRL during training on multifidelity NAS environments. See text for description of the two multifidelity scenarios (i) and (ii). In both cases, $\#\tau^{\text{lo}} = 5/(\mathcal{T}(s^{\text{hi}}), a))$.

ronments. For both cases, we train both our proposed MFMCRL and the MCRL exactly as we did in Section 4.1, and we report the mean and standard deviation of test episode rewards in Figure 3. We can observe that our multifidelity RL framework does indeed improve over standard RL and that performance gains are higher when the low- and high-fidelity environments are more similar, case (i).

## 5 Conclusion

In this paper we have studied the RL problem in the presence of a low- and a high-fidelity environment for a given control task, with the aim of improving the agent's performance in the high-fidelity environment with multifidelity data. We have proposed a multifidelity estimator based on the method of control variates, which uses low-fidelity data to reduce the variance in the estimation of the state-action value function. The impacts of variance reduction on policy improvement and policy evaluation are theoretically analyzed, and a multifidelity Monte Carlo RL algorithm (MFMCRL) is devised. We show that for a finite budget of high-fidelity data, the MFMCRL agent can well exploit the cross-correlations between low- and high-fidelity data and yield superior performance. In our future work, we will study the design of a control-variate-based multifidelity RL framework with function approximation to solve continuous state-action space RL problems.

## 6 Broader impact

*Positive impacts:* The energy/cost associated with generating low-fidelity data is generally much smaller than that of high-fidelity data. By leveraging low-fidelity data to improve the learning of RL agents, greener agents are realized. *Negative impacts:* Running multifidelity RL agent training with weakly-correlated low- and high-fidelity environments can be wasteful of resources since the benefits in this case are not significant.

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
