# OpenReview forum: "Multifidelity Reinforcement Learning with Control Variates"
_NeurIPS.cc/2022/Conference — NeurIPS 2022 Submitted_

### Official Review · Reviewer_rJ7u · 2022-07-03

**Rating:** 4
**Confidence:** 4
**Soundness:** 3 good
**Presentation:** 3 good
**Contribution:** 2 fair

**Summary:**

The paper is framed in the setting of multi-fidelity Reinforcement Learning (RL), i.e., the setting in which multiple instances of the environment are available, each with a different accuracy on the reward function and transition dynamics. Specifically, the paper considers the presence of two instances (an exact one and an approximate one) and proposes an approach to effectively exploit this opportunity to improve the estimation of the value functions. In particular, by assuming that the reward random variable of the two environments are correlated, a suitable control variance is exploited to reduce the variance of the estimate and achieving smaller sample complexity. A theoretical analysis is provided, showing results regarding the accuracy of the value function and of the greedy policy. An experimental evaluation on both synthetic domains and on a NAS (neural architecture search) is provided, highlighting the advantages of the proposed approach.

**Questions:**

I would appreciate it if the authors could clarify the issues [About Correlation of the Rewards] and [Estimation of Q^{lo}_{\pi}]. Furthermore, I add some minor questions:

- In Section 2.1, the authors require the state and action spaces to be finite. Is this a crucial assumption? Can the approach be employed in continuous state-action spaces?

- In real-world applications of multi-fidelity RL, is it reasonable to assume that the mapping function from high-fidelity states to low-fidelity states $\mathcal{T}$ is known?

- Is the assumption that the low-fidelity environment admits a generative model-kind of interaction crucial for the approach? On the one hand, I guess it is because this is exploited for estimating the true value of  $Q^{\text{lo}}_{\pi}$. Can the authors elaborate?

**Limitations:**

The limitations and impact of the paper are properly addressed by the authors.

**Strengths And Weaknesses:**

- The main strong point of the paper is the fact that it addresses a very relevant topic, i.e., multi-fidelity RL, that emerges in several real-world applications.

- [Novelty] The approach proposed in the paper is novel per se, although it can be considered an application of the control-variate techniques to the multi-fidelity setting. The theoretical evaluation is novel and succeeds in showing some advantages of the proposed approach (although I have some concerns that I will detail below)

- [About Correlation of the Rewards] The crucial assumption that makes the proposed approach work is that the reward random variables collected in the high-fidelity and low-fidelity environments in the same state are correlated in a statistical sense (i.e., with a non-zero correlation coefficient). This is quite different from the usual idea of fidelity in multi-armed bandits (e.g., "Kandasamy, Kirthevasan, Gautam Dasarathy, Barnabas Poczos, and Jeff Schneider. "The multi-fidelity multi-armed bandit." Advances in neural information processing systems 29 (2016).") in which the fidelity is represented as a bias on the expected reward (so, it is not a correlation in the statistical sense). Consequently, in order for the approach to make sense, it must be that the reward is stochastic given the current state and action. It seems that the approach cannot be applied to rewards that are deterministic functions of the state-action pair. Indeed, in such a case, there would be no correlation. Requiring correlated reward random variables seems quite restrictive in my view. Can the authors elaborate on this point?

- [Estimation of $Q^{lo}_{\pi}$ ] As the authors acknowledge, in order to compute the control variate, it is necessary to have access to the value of the true low-fidelity value function Q^{lo}_{\pi}. This is not available in practice. The authors propose to collect a large number $m$ of return samples from the low-fidelity environment. Although $m$ can be chosen to be large (assuming that the cost of collecting samples in low-fidelity is negligible), it represents a further source of uncertainty that will impact the computation of all the relevant quantities. Moreover, in the control variate, the covariances and the correlation coefficient are estimated from samples as well. Are these further sources of uncertainty accounted for in the theoretical analysis of Section 3.3?

*Overall*
The paper has some novelty and the experimental results, although limited to one realistic environment (and some synthetic ones), show some advantages of the proposed approach, I think the paper is currently borderline. I would appreciate it if the authors could clarify the concerns about the formulation of the fidelity (i.e., the correlation between the random variables).

---

> ### Author Response · Authors · 2022-08-02
> **Response to reviewer questions**
>
> We would like to thank the reviewer for their great efforts in reviewing our work. We will revise the manuscript based on these comments in the camera ready version. Meanwhile, we provide our responses to address the concerns.
>
> **About Correlation of the Rewards**
>
> For the proposed approach to work, the high-fidelity return, rather than the rewards, in the high-fidelity environment needs to be correlated with that in the low-fidelity environment. By definition, the returns are the cumulative discounted rewards computed over many time steps. The distribution of a return from a given state-action pair (s,a) depends on the transition function P(s’|s,a), reward function R(s,a), and the policy function \pi(a|s). If one or more of these three functions is stochastic, then the return from a pair (s,a) will be a random variable with a non-zero variance. Accordingly, the estimation of the mean of a return (Q-function) from a pair (s,a) will also have some variance, and our approach is beneficial in the way it reduces estimation variance by leveraging cross-correlations between low- and high-fidelity returns. Therefore, the only case in which our approach does not make sense is when the MDP admits a deterministic transition function, a deterministic reward function, and the agent follows a deterministic policy, in which case the Q-function is deterministic and can be estimated using a single rollout from a given (s,a) pair. This case however is rarely encountered in practice, because even in simple deterministic environments (where both the transition function and rewards are deterministic), the policy is often an exploratory stochastic policy and so returns will be random variables. We hope that the distinction between correlated rewards and correlated rewards addresses your concern.
>
> **Estimation of $Q^{lo}_{\pi}$**
>
> We agree that estimating the Q-function of the low-fidelity environment introduces uncertainty, although this uncertainty can be controlled (and minimized) by increasing m, the number of low-fidelity return samples. As we have mentioned in the manuscript, sampling cost is application dependent, and it is up to the practitioner to assign cost and determine low- and high-fidelity sampling budgets. In general, our proposed approach is mostly beneficial in applications where low-fidelity samples are much cheaper (e.g. simulators) relative to high-fidelity samples (e.g. samples from laboratory apparatus). The numerical results of Figs. 2(a) and 4(a) of the manuscript show that performance improvements can be achieved by increasing m, which supports our argument that the uncertainty looming around estimating the low-fidelity Q-function and correlation coefficients can be effectively controlled.
>
> Regarding the second part of your point, our theoretical analysis is mainly provided to expose the impacts of variance reduction on the two fundamental tasks in RL, which are policy evaluation and policy improvements, and to demonstrate that our proposed method is theoretically sound. As such we assume the true covariances and correlation coefficient to maintain mathematical tractability in our analytic derivations.
>
> **Answers to Questions**
>
> 1. Extending our proposed approach to continuous state-action spaces is possible in theory, however, it requires careful treatment of approximation errors which will be introduced when function approximators are used to represent the Q-function. This extension is a very exciting direction which we are actively pursuing in our future work.
>
> 2. T is problem-specific and typically computational modeling scientists know how the state space in different fidelity levels can be mapped. For instance, a low-fidelity simulator of a robot might ignore some physical quantities (setting those variables to 0, for instance), which yields the many-to-one map from high-fidelity states to low-fidelity states. As another example, S might be a discretization of some R^d in some applications, i.e. a grid. In this case, S^hi may represent a fine grid while S^lo may represent a coarse grid, and T can be a simple distance-based map. Please notice that we do not require knowledge of the map that exists between dynamics models in the low- and high-fidelity environments, which are much harder to capture in practice.
>
> 3. Having a generative low-fidelity environment makes things easier for our proposed learning approach as the agent needs to estimate the low-fidelity Q-function for the low-fidelity pairs (s^low=T(s^hi), a) that correspond to (s^hi, a) pairs encountered in the high-fidelity environment. With a generative low-fidelity environment, the agent can start the MC rollout/simulation from any desired pair, rather than having to keep interacting with the low-fidelity environment until a desired pair is encountered (which is less sample efficient).

---

> > ### Comment · Reviewer_rJ7u · 2022-08-08
> > **Post-Rebuttal Feedback**
> >
> > I thank the authors for the feedback. I have read it together with the other reviews.
> >
> > I think that my concerns, apart from [About Correlation of the Rewards], are properly addressed by the authors. However, the one about the [Estimation of $Q^{lo}_{\pi}$ ]  and its role in the analysis represents a limitation that should be properly discolosed in the paper.
> >
> > For what concerns my issue [About Correlation of the Rewards], I think the authors' response is not satisfactory. My concern is about the fact that the assumption is that low-fidelity returns (not reward, as I was previously mentioning) are statistically correlated with high-fidelity returns. The authors, in the rebuttal, state that "the only case in which our approach does not make sense is when the MDP admits a deterministic transition function, a deterministic reward function, and the agent follows a deterministic policy". However, one can easily imagine settings where the transition function, the reward, and the policy are stochastic, and still low-fidelity and high-fidelity samples are not correlated. What if the high-fidelity and low-fidelity environments generate statistically independent trajectories? They can be arbitrarily similar in distribution (they can even be identically distributed), but this is not relevant. What is relevant for the analysis of Section 3.2 is the statistical correlation between the samples. If such a correlation (in the statistical sense) is not present, the whole approach loses its purpose.
> >
> > This represents, at present, a severe limitation of the work since assuming a statistical correlation between low and high-fidelity environments seems to me quite unrealistic, and the authors did not motivate it.

---

> > > ### Author Response · Authors · 2022-08-08
> > > **Comment on reviewer's post-rebuttal feedback**
> > >
> > > We would like to thank the reviewer for reading our response and for their feedback. We do agree that having statistical correlation between low- and high-fidelity returns given a state-action pair is necessary for our approach to deliver the desired gains, and that if the correlation is absent, there will be no gain. However, we would like to stress that we are studying a problem setup where the low- and high-fidelity environments are related to each other by design such that correlation is present. Specifically, we study the case where the low-fidelity environment is a cheap imperfect simulation of the high-fidelity environment. The high-fidelity environment can be a real-world environment or a very expensive simulation-based environment. In this problem setup, the low-fidelity environment is designed to capture some of the intricacies of the high-fidelity environment, albeit not perfectly, and the task is to leverage this cheaply available data from the low-fidelity environment to learn a better policy in the high-fidelity environment. Example real-world application areas for this setup include computational physics, geosciences, and engineering applications based on partial differential equation models [a,b] such as molecular dynamics and computational fluid dynamics (weather) simulations. We will revise the introduction and the problem setup in 3.1 to better state the motivation with real-world computational physics applications.
> > >
> > > [a] Benjamin Peherstorfer, Karen Willcox, and Max Gunzburger. "Survey of multifidelity methods in uncertainty propagation, inference, and optimization." SIAM Review 60, no. 3 (2018): 550-591.
> > >
> > > [b] Xuhui Meng and George Em Karniadakis. "A composite neural network that learns from multi-fidelity data: Application to function approximation and inverse PDE problems." Journal of Computational Physics 401 (2020): 109020.

---

### Official Review · Reviewer_HcCy · 2022-07-09

**Rating:** 7
**Confidence:** 4
**Soundness:** 3 good
**Presentation:** 3 good
**Contribution:** 4 excellent

**Summary:**


The paper presents a new methodology for multi-fidelity RL by leveraging low-fidelity data to decrease the variance in the MC-estimates of the value function.  The new algorithm, MFMCRL, leverages theory from control-variates to decrease the variance based on the observed correlation fbetween the high and low value returns.  This is a different approach than existing MFRL techniques that choose which simulator to gather data from and need bounds on the relationships between the simulators.  Theoretical results quantifying the improvements from this approach in both policy evaluation and policy improvement are derived.  Empirical results are presented with synthetic MDPs and in a NAS (hyperparameter tuning) environment.


**Questions:**

Can the authors provide a more precise description of the i != j covariance construction and why the indexes should match up?

Why is the analysis restricted to only domains with terminal states?

Can the authors comment on the dimensions [(1) (2) and (3) in the final paragraph above] where the new algorithm may not be as strong as earlier MFRL methods?

**Limitations:**

As mentioned above, while I think the new approach has a benefit of requiring less strict knowledge about the relationship between low and fidelity environments, there seem to be areas where the new approach is not as strong as prior approaches.  These specific areas are (1) accommodating more than 2 environments, (2) losing sample efficiency by learning from MC returns rather than doing model based learning, and (3) not actively choosing which environment to sample, which will not be as efficient in querying the expensive high-fidelity environment.  If these limitations hold with respect to prior work they should be mentioned in the paper.

**Strengths And Weaknesses:**

Overall I think this is a very clever idea and a nicely scoped paper.  The approach is novel compared to existing MFRL techniques in that it takes a very different approach to gathering data.  There are places though, where I think the paper needs to be more precise in its terminology and also point out some disadvantages of the current formulation with respect to prior MFRL work.  Details below:

More precise terminology needed:
The assumption of independence at line 110 (when i != j) and the related covariance construction in (11) are very confusing to me.  I and j here are indexing two different sets (the high fidelity and low fidelity one) from what I can tell where each element is the return of a trajectory in the related MDP.  So there are n trajectories from Lo and n from high, right?  In that case, isn’t there likely non-zero covariance between lots of these trajectories?  That is, a rollout of the same policy in these two similar MDPs will produce correlated returns – if half of them take a right turn (stochastically) then we should expect 50% of the trajectories in Z to be correlated with another 50% in W.  So why would we expect 0 correlation just because the indexes don’t match?  What is special about two matching indexes (Z_i and W_i)?  Can the authors provide a concrete example showing why we would expect i != j to be important?

I was surprised the environment setup in Section 2.1 requires an episodic domain and specifically one with terminal states.  Is this necessary for the algorithm / analysis because it is based on MC sampling and are there natural extensions to non-episodic domains.

The sentence starting on line 120 could use a citation.

Line 199 – A  note should be made here about why the new approach needs to use (inefficient) MC returns instead of utilizing bootstrapping in learning the value function.  I presume since the latter is a biased estimator it is not usable in this framework.  That is fine, but the point should be made directly here.

I think the pseudocode block from the appendix needs to be in the main paper.  The current description of the algorithm in the main paper is not really usable without it.  If this paper gets accepted I suggest using the extra page to include that.

On potential disadvantages with regards to previous methods:
I agree with the authors that the new approach has fewer assumptions about prior knowledge compared to previous MFRL methods. Being able to learn the correlations between the environments is a nice improvement.  But there are other dimensions where the current approach is not as efficient or flexible as previous MFRL approaches and those should be called out directly in the related work comparison.  Specifically (1) the current work always assumes there are only 2 environments (low and high) while previous work considered any number of environments – can the new framework accommodate multiple low-fidelity environments?  If so or if not this should be mentioned. (2) The new framework is doing value-based RL using MC returns, which is likely going to be very inefficient compared to the model based approaches used in prior work and (3) The authors make a point about not needing to use more complicated exploration strategies to decide what environment to sample from, but that will lead to inefficiencies.  The current approach will sample from both Low and High on every iteration, which could be very costly, whereas previous MFRL approaches only sample from High when they feel they are “ready” to take on the high-fidelity environment.  So yes, the new approach will be more sample efficient than only sampling from High, but it seems it will be much less sample efficient than previous MFRL approaches in terms of sampling the highest fidelity environment.

---

> ### Author Response · Authors · 2022-08-02
> **Response to reviewer questions**
>
> We would like to thank the reviewer for their great efforts in reviewing our work. We will revise the manuscript based on these comments in the camera ready version. Meanwhile, we provide our responses to the comments raised by the reviewer.
>
> **Regarding precise terminology**
>
> Thank you for your question and we apologize for this confusion. First of all, let us focus on the high-fidelity environment. It is worth noting that for the purpose of estimating mean returns (or the Q-function), each rollout/trajectory sample (i.e. a sequence of state-action-reward-next states tuples) is generated by interacting with the “high-fidelity” environment based on a policy which selects actions a at state s with probability pi(a|s). For a given state-action pair (s_0,a_0), the environment evolves to the next states s’ with probability P(s’|s,a). A rollout/trajectory which starts at (s_0,a_0) and terminates at s_T, yields the sequence of rewards r_1, r_2,..., r_{T-1}, which can be used to compute one sample i (G_i) of the long-term discounted returns for the pair (s_0,a_0). Please note that we use the first-visit monte carlo estimator which is unbiased. The n rollouts/trajectories (with indices i=0,1,...,n-1) that start at (s_0,a_0) are independently generated, but identically distributed (i.i.d), and so any two return samples (G_i,G_j) where (i != j) are uncorrelated.
>
> Next, the agent maps each “high-fidelity” rollout/trajectory (indexed by i ) with high-fidelity return G_i^hi, to  the corresponding “low-fidelity” rollout/trajectory (as explained on line 203 in the manuscript and line 4 in the algorithm), and then computes the low fidelity return G_i^low. By construction, G_i^hi and G_i^lo are correlated random variables, having been computed from the same realization of the random process over (s,a). In summary, returns computed based on different rollouts/trajectories (i != j) are iid and uncorrelated. High and low fidelity returns computed based on the same trajectory, are correlated. We hope that this explanation clears the confusion.
>
> **Environment setup**
>
> In our proposed approach, we use the first-visit monte-carlo sample average to estimate the mean returns or Q-function, and this estimator is known to be unbiased for episodic domains with terminal states (Sutton and Barto). Using this estimator in a non-episodic domain, while perhaps viable, leads to a biased estimator, although the bias can be decreased by assuming a very long horizon and/or using a discount factor <1. Our analysis/algorithm can be employed in a non-episodic domain if the estimator bias is tolerable.
>
> **Line 199, MC returns**
>
> Yes, you are absolutely right. Incorporating this framework within bootstrapping/temporal difference methods requires careful treatment of the bias introduced by such estimators, which we are currently looking at in our follow up work.
>
> **Dimensions 1,2,3**
>
> 1. The proposed method can readily be extended to the case of more than two environments, but for clarity and ease of presentation, we ought to focus on the two-environment case. In general, if there is a chain of K environments with k=K being the highest fidelity environment and k=1 is the lowest fidelity environment, the multi-fidelity estimator would look like (https://postimg.cc/hQvJfhKv). Basically, we can use the correlation that exists between level k \in {1,2,..,K-1} and K, to reduce the variance in the estimation at level K  using control variates.
>
> 2. In our work, we consider a general MDP setting with a stochastic transition function. Learning a model in such a setting is much harder compared to the case of MDPs with a deterministic transition function (which has been the main focus of previous works [1]).
>
> 3. In RL, multi-fidelity data can be exploited in many stages of the decision-making pipeline, including estimation, exploration, and exploitation. Previous approaches, where the agent samples from a high-fidelity environment only when it feels ready to take on it, essentially use multi-fidelity data to manage exploration and exploitation. The most tricky part here is the quantification of “readiness.” Existing works typically resort to heuristics whereby hyper-parameters that dictate when the agent should switch up or down the environment chain are introduced (e.g. hyper-parameters to compensate for the difference between the low and high fidelity Q-functions). Sample efficiency of these methods depends to a great extent on these hyper-parameters. Tuning of these hyper-parameters is problem specific and can be challenging unless more is known about how the high and low-fidelity environments are related. Nevertheless, we do agree that this adaptive sampling can be efficient, and this is why we view our proposed approach, which introduces a new method to improve estimation with multi-fidelity data, to be complementary to existing approaches and can be used in conjunction with them as mentioned on line 144.

---

> > ### Comment · Reviewer_HcCy · 2022-08-08
> > **author response acknowledgment**
> >
> > I have read the author response and it clarified the algorithm itself.  I had not realized that the low-fidelity environments were not used for simulation, but instead for state embedding.  I again caution the authors that their treatment of prior work needs more nuance.  Saying that previous work like [1] focused on deterministic domains makes it sound like the new paper is the first to deal with stochasticity but that is not the case.  For instance [9] dealt with the general MDP case.  Similarly, while other algorithms had hyper-parameters, so does the new algorithm, and some of the older ones had strong  (albeit parameter-specific) guarantees.  There are advantages and disadvantages to the new approach, and that is fine.  Anyway it looks like the reviewers will have plenty to discuss but I have all I need from the authors now.  thank you for your time clearing up the misunderstandings.

---

> > > ### Author Response · Authors · 2022-08-08
> > > **Comment on reviewer's acknowledgement**
> > >
> > > We would like to thank the reviewer for reading our response. In our algorithm, the low-fidelity environment is used to provide two pieces of information for any given state-action pair. First, we use it to map high-fidelity trajectories to low-fidelity trajectories, which allows us to compute  $\hat{Q}_\pi^\text{lo}(s^\text{lo},a)$; a noisy sample mean estimate of low-fidelity returns. Second, the low-fidelity environment is used to simulate a large number of trajectories from $(s^\text{lo},a)$ so that we have a better estimate of the population mean ${Q}_\pi^\text{lo}(s^\text{lo},a)$. Together with correlation estimates, these quantities are used in the control variate framework to design a reduced-variance multi-fidelity estimator for ${Q}_\pi^\text{hi}(s^\text{hi},a)$, which is the quantity we care about as it encodes the policy in the high-fidelity environment. Regarding the treatment of previous works, we agree with the reviewer on these points and will take utmost care when we revise the related sections in the camera ready version.

---

### Official Review · Reviewer_8Vzs · 2022-07-10

**Rating:** 6
**Confidence:** 3
**Soundness:** 3 good
**Presentation:** 3 good
**Contribution:** 2 fair

**Summary:**

The authors focus on improving the performance of reinforcement learning algorithms in the presence of multi-fidelity data through the use of control variates.


**Questions:**

## Preliminaries and related work
1. "It is well known that there exists at least one optimal policy π∗ such that Vπ∗ (s) = max π Vπ (s), ∀s ∈ S and Qπ∗ (s, a) = max π Qπ (s, a), ∀s, a ∈ S × A [ 2]. Furthermore, a deterministic policy that selects the greedy action with respect to Qπ∗ (s, a), ∀s ∈ S, is an optimal policy." -- If you make statements please include a citation. I believe that a good reference for this statement would be (1).
2. "Applications of the method of control variates extend beyond variance reduction. For example, the concept of control variates is used in [ 27 ] to design a fusion framework to combine an arbitrary number of surrogate models optimally." --Also the notion of a baseline in reinforcement learning has been well studied since the inception of policy gradient methods (2). The literature surrounding baselines should be mentioned if you are talking about variance reduction in RL methods (3,4).
3. "In [1], a policy search algorithm is proposed that leverages a crude approximate model ˆP of the true MDP to quickly learn to perform well on real systems." -- Can you state explicitly the policy search algorithm that is used.
4. "From an optimization viewpoint, this approach is reasonable only if the lower-fidelity value function lies in the vicinity of the optimal high-fidelity value function, a situation that cannot be guaranteed or known a priori in general." -- can you make this more explicit? Do you mean that it is close in terms of the true value function under co-simulation between low-fidelity and high-fidelity models?
5. It seems like this work is deeply related to asymmetric learning algorithms (5,6). Can you comment on the connection to this sub-field and how it relates to your work? It seems as though your analysis might benefit from viewing the low-fidelity process as a POMDP, and the high-fidelity process as the true underlying sequence of states.

## Multifidelity estimation in RL

1. "High-fidelity environments usually capture more state information than do low- fidelity environments so T can be a many-to-one map" -- It seems like this would imply that the state-space would be implicitly or explicitly larger, which you would imagine might actually slow down learning. Can you comment on this?
2. "Notice that G hi and G lo are correlated r.vs. in this multifidelity setup" -- This is only true for the first step of the monte-carlo rollout of the low fidelity simulator though. Meaning that for t > 0 all rewards could be de-correlated and the level of correlation of G should be dependent on the discount factor which close to 1 becomes vanishingly small.  Is this correct?
3. Equation 7 implies that you are able to exactly estimate Q in the low-fidelity simulator which seems unrealistic in general. Is this the case, or did I misunderstand the notation?
4. Please include a reference for the policy improvement theorem in 3.3.2 as it does have some requirements that I am unsure are met in your setting. Depending on the variant (e.g. are we looking at exact policy iteration, asynchronous policy iteration or approximate policy iteration?
5. " This assumption basically ensures that all actions at the target state shi have been explored equally well and enables us to make fair comparisons about estimator performance." -- This statement seems inconsistent with the message expressed earlier which was that your algorithm in fact addresses the exploration issue in systems where correlation in the expected reward ahead is high between high and low fidelity simulators.

## Numerical experiments
1. Where are other common baselines for the neural architecture search? I appreciate the need to compare over the non-variance reduced baseline, but it seems like having examples from other algorithms in the paper is necessary to see exactly where this technology stands with respect to the larger NAS literature.
2. What were the relative clock-times of NAS with and without the control-variate schemes? Were they comparable?
3. The plot for figure 3 seems incorrect as for the non-variance reduced algorithm the standard deviation does not fully overlap the mean, are you sure you did not report the mean and two quantiles?

## Citations mentioned
(1) Bertsekas, Dimitri. Reinforcement learning and optimal control. Athena Scientific, 2019.

(2) https://people.cs.umass.edu/~barto/courses/cs687/williams92simple.pdf

(3) https://proceedings.neurips.cc/paper/2010/file/35cf8659cfcb13224cbd47863a34fc58-Paper.pdf

(4) https://arxiv.org/pdf/1506.02438.pdf

(5) https://arxiv.org/abs/1710.06542

(6) https://arxiv.org/abs/2012.15566

**Limitations:**

The major weakness mentioned by the authors is the setting when the correlation between low fidelity and high-fidelity simulation is low their variance reduction algorithm can actually perform worse then the baseline. I again think that the authors need to include other baselines to give an idea of how performant their algorithm is by comparison. It need not beat these other algorithms, but other standard baselines (especially on NAS) should be included.

**Strengths And Weaknesses:**

## General Comments
I really like this paper, and hope that it gets accepted.  However there are a few, hopefully easy, problems that the authors need to address. First the literature review needs to be expanded to include background from variance reduction in classic RL, and asymmetric RL. Second, the assumptions and discussion surrounding policy iteration need to be more explicit in the main paper, or pointed to in the appendix. Lastly, they need to include, potentially in the appendix, other baseline algorithms in the NAS example. If these problems can be addressed in the rebuttal  period, I will increase my score.

## Strengths
1. This is actually a really clearly written paper that I enjoyed reading.
2. I think that the area that it considers is an important one.
3. The experiments are clearly presented, and the weaknesses of the work, as well as their strengths are discussed in detail.

## Weaknesses
1. The authors seem to ignore the full literature surrounding the learning of baselines in general reinforcement learning which I think does not set the right context for why the question the authors answer is interesting.
2. Also ignores recent work in asymmetric reinforcement learning, which I think would benefit the larger story and analysis. Especially in connecting, in a more principled way, low fidelity and high-fidelity simulation.
3. The NAS experiments do not include other baseline algorithms. Though I am not familiar with this specific benchmark, I have no doubt there are other algorithms which should be included to give a point of reference for what the gap between RL based methods and other methods is.
4. The authors seem as though they are not being fully transparent about the types of MDPs for which this algorithm can currently be configured with. Does it only work with low-dimensional state-action spaces and toy problems, or can it be configured with more difficult continuous control tasks?
5. The discussion and assumptions surrounding the policy iteration alluded to in section 3.3 was basically non-existent. For better clarity it seems like a more explicit discussion should be included (either in the main paper or the appendix).

---

> ### Author Response · Authors · 2022-08-02
> **Response to reviewer questions**
>
> We would like to thank the reviewer for their great efforts in reviewing our work. We will revise the manuscript based on these comments in the camera ready version. Meanwhile, we provide our responses to the questions raised by the reviewer.
>
> **Preliminaries and related work**
>
> [Q1,Q2,Q5]: The key distinction between variance reduction in policy gradient methods using baselines and our work is in the scope: 1) we focus on value-based RL, 2) we study the problem of learning a policy that is optimal with respect to the high-fidelity environment by collecting additional cheap data from low-fidelity environments.
>
> In asymmetric RL, additional information is available to the agent during training that is not available at test time. During training, the agent has access to full state information as well as to partial observations, while at testing the agent only has access to partial observations. The reward functions and transition dynamics are the same during training and testing. In our work, the low-fidelity and high-fidelity environments are modeled by two different MDPs with different (but related) state spaces, reward functions, and transition functions.
>
> Q3. Differential dynamic programming.
>
> Q4. We mean that the gap between the optimal low-fidelity value function (w.r.t to low-fidelity environment) and the optimal high-fidelity value function (w.r.t. to high-fidelity environment): Q^*_hi(s,a) - Q^*_lo(s,a), is not too large and bounded. In this case, transfer of the policy from the low to the high environment can produce a good performance with a bounded optimality gap.
>
> **Multifidelity estimation in RL**
>
> Q1. That is true, however  this challenge is inherent to RL (curse of dimensionality). In (4.2), we have studied whether having access to a low-fidelity environment with a smaller state space can benefit learning in the high-fidelity environment using our proposed framework. The results of Fig. 3 are encouraging as it shows that learning in the high-fidelity environment (MCRL, blue curve) can be sped up by using data from a low-fidelity environment (MFMCRL case ii, red curve).
>
> Q2. This is not necessarily true. The distribution of a return R.V. from a state-action pair depends on the stochastic transition function, reward function, and policy. In a multi-fidelity setting, the level of correlation between G_hi and G_low depends on how the low-fidelity transition and reward functions relate to the high-fidelity transition and reward functions, respectively. In (4.1), we consider low-fidelity transition and reward functions which are noisy versions of their high-fidelity counterparts. As we can see in Figs. 2b & 2c, higher correlations are obtained when the low-fidelity environment is less noisy, which yields higher performance gains.
>
> Q3. That is true in equation 7, but in lines 231-233 we mention that the estimation of Q_low can be performed in practice by collecting a large number of samples from the cheap low-fidelity environment to reduce uncertainty. In Fig. 2a, we show that the performance of the agent improves by increasing the number of low-fidelity samples.
>
> Q4. Since our algorithm proposes a new estimator for the Q-function but does not change approximate policy iteration otherwise, the convergence of our algorithm is guaranteed under the same conditions that guarantee convergence for the on-policy first-visit MC control algorithm with ϵ-soft policies.
>
> Q5. Performance gains achieved by our algorithm are purely a result of reducing the variance in the estimation of the Q-function using our multi-fidelity estimator. Our algorithm does not modify the exploration process, as stated in lines 55, 56-58, 142-145.
>
> **Numerical experiments**
>
> Q1. In (https://postimg.cc/svXm98Qt), we plot the 95th percentile of the validation accuracy (optimization objective in NAS) of the architectures evaluated by aging evolution, RL, and MFMCRL. Aging evolution is considered one of the strong baselines in NAS, outperforming RL (Figure 3 in [a], and Figure 6 in [b]). Aging evolution is stateless (decision variables do not depend on a state), and focus directly on finding high-performing architectures. In RL, the agent’s task is to learn a policy for configuring the nodes of an architecture given its state and the hope is to discover high-performing architectures as a by-product of stateful policy learning. That said  and given the high sample complexity of RL, RL-based algorithms tend to underperform in NAS.
>
> [a] arXiv:1802.01548
> [b] arXiv:2001.00326
>
> Q2. For all architectures for which the high-fidelity reward is evaluated, the corresponding low-fidelity reward will be available at no additional cost in NAS. By design, our proposed algorithm may also need to sample other low-fidelity rewards as it estimates Q-low. Hence, it is expected that our proposed multi-fidelity scheme will run slower than the original scheme.
>
> Q3. In figure 3, we report the mean test episode rewards plus/minus standard deviation.

---

> > ### Comment · Reviewer_8Vzs · 2022-08-09
> > **Response**
> >
> > Thanks for answering some of my questions. Regarding [Q1,Q2,Q5], after getting information on what algorithm was being used (DDP) the distinction between variance reduction in your setting policy gradient style methods is more clear. My only comment on the Asymmetric information line of work was that making an explicit modeling assumption about the relationship between the two MDPs might be beneficial. I am also still somewhat unconvinced by the response for multi-fidelity estimation [Q2], as the possibility of my statement being true seems somewhat concerning. That said, I still enjoyed reading the paper, and it seems as though the authors worked hard to ensure the paper was as clear as possible so I will increase my score a bit.

---

### Meta-Review · Area_Chair_xBW9 · 2022-09-13

**Recommendation:** Reject
**Confidence:** Less certain

**Metareview:**

All reviewers agree on the novelty and significance of the topic addressed by the paper and feel that the paper is at the borderline of being ready for publication. But in the end, as visible in the lengthy comments and discussions about the paper, there are too many details and clarifications missing in this paper to push it solidly across the acceptance threshold. Given all the technical discussions, the authors should be in an excellent position for a significantly improved resubmission in the future.

**Award:**

No

---

### Decision · Program_Chairs · 2022-09-14

Reject